# Efficacy and safety of pyronaridine-artesunate (PYRAMAX) for the treatment of *P. falciparum* uncomplicated malaria in African pregnant women (PYRAPREG): study protocol for a phase 3, non-inferiority, randomised open-label clinical trial

Moussa Djimde,[1] Japhet Kabalu Tshiongo ,[2] Hypolite Mavoko Muhindo,[3] Halidou Tinto,[4] Esperanca Sevene,[5,6] Maminata Traore,[7] Anifa Vala,[8] Salesio Macuacua,[9] Berenger Kabore,[7] Edgard Diniba Dabira,[10] Annette Erhart,[10] Hamadoun Diakite,[1] Mohamed Keita,[11] Mireia Piqueras,[12] Raquel González ,[12] Clara Menendez,[12] Thomas PC Dorlo,[13,14] Issaka Sagara,[1] Petra Mens,[15] Henk Schallig,[15] Umberto D'Alessandro,[16] Kassoum Kayentao [1]

MD and JKT contributed equally.

For numbered affiliations see end of article.

**Correspondence to**
Dr Kassoum Kayentao;
kayentao@icermali.org

## ABSTRACT

**Introduction** Malaria infection during pregnancy increases the risk of low birth weight and infant mortality and should be prevented and treated. Artemisinin-based combination treatments are generally well tolerated, safe and effective; the most used being artemether-lumefantrine (AL) and dihydroartemisinin-piperaquine (DP). Pyronaridine-artesunate (PA) is a new artemisinin-based combination. The main objective of this study is to determine the efficacy and safety of PA versus AL or DP when administered to pregnant women with confirmed *Plasmodium falciparum* infection in the second or third trimester. The primary hypothesis is the pairwise non-inferiority of PA as compared with either AL or DP.

**Methods and analysis** A phase 3, non-inferiority, randomised, open-label clinical trial to determine the safety and efficacy of AL, DP and PA in pregnant women with malaria in five sub-Saharan, malaria-endemic countries (Burkina Faso, Democratic Republic of the Congo, Mali, Mozambique and the Gambia). A total of 1875 pregnant women will be randomised to one of the treatment arms. Women will be actively monitored until Day 63 post-treatment, at delivery and 4–6 weeks after delivery, and infants' health will be checked on their first birthday. The primary endpoint is the PCR-adjusted rate of adequate clinical and parasitological response at Day 42 in the per-protocol population.

**Ethics and dissemination** This protocol has been approved by the Ethics Committee for Health Research in Burkina Faso, the National Health Ethics Committee in the Democratic Republic of Congo, the Ethics Committee of the Faculty of Medicine and Odontostomatology/Faculty of Pharmacy in Mali, the Gambia Government/MRCG Joint Ethics Committee and the National Bioethics Committee for Health in Mozambique. Written informed consent will be obtained from each individual prior to her participation in the study. The results will be published in peer-reviewed open access journals and presented at (inter)national conferences and meetings.

**Trial registration number** PACTR202011812241529.

## STRENGTHS AND LIMITATIONS OF THIS STUDY

⇒ This is a randomised clinical trial that will provide evidence on the safety and efficacy of a newly registered artemisinin-based combination, pyronaridine-artesunate (PA), for the treatment of uncomplicated malaria in African pregnant women during the second and third trimesters of pregnancy.

⇒ The trial will generate additional data on the safety and efficacy of dihydroartemisinin-piperaquine and artemether-lumefantrine and will increase substantially the treatment options for malaria in pregnancy.

⇒ As PA is the last artemisinin-based combination to have been registered for the treatment of malaria, the proposed trial will complement the information already available for malaria treatment in children and non-pregnant adults.

⇒ Weaknesses include the lack of regular monitoring of children within the first year of birth and the non-inclusion of infected women in the first trimester.

## INTRODUCTION

Pregnancy is associated with hormonal changes and reduced immunity that make pregnant women more vulnerable to malaria.[1]

Malaria during pregnancy increases the risk of low birth weight (LBW), which is associated with higher infant mortality.[2] The prevention and control of malaria during pregnancy is based on insecticide-treated nets, intermittent preventive treatment during pregnancy (IPTp) with sulfadoxine-pyrimethamine (SP) and effective case management of clinical malaria, including anaemia.[3] The latter relies on artemisinin-based combination therapies (ACTs) which are well tolerated, safe and efficacious.[4] Their use is currently limited to the second and third trimesters, although the risk of miscarriage, stillbirths or major birth defects when used during the first trimester is better than quinine, the recommended drug.[5 6]

Currently available ACTs for the treatment of *Plasmodium falciparum* malaria during the second and third trimesters of pregnancy, namely artemether-lumefantrine (AL), artesunate-amodiaquine, mefloquine-artesunate and dihydroartemisinin-piperaquine (DP), are generally safe and efficacious.[4] Nevertheless, when considering efficacy and tolerability, DP is probably the best option as it provides long post-treatment prophylaxis and is well tolerated. DP is also considered for IPTp as a possible alternative to SP, where resistance to this treatment is high.[7] Its deployment as IPTp would significantly limit its use as a curative treatment. There is a need to increase the therapeutic options to treat malaria during the second and third trimesters of pregnancy. Pyronaridine-artesunate (PA) may be a possible candidate. PA tablets and granules (Pyramax) have received a positive European Medicines Agency/Article 58 scientific opinion for the treatment of acute uncomplicated *P. falciparum* or *Plasmodium vivax* malaria.[8] The treatment (tablets) is currently registered in 19 African countries and five Asian countries. Pyramax (granules) is registered in 15 African countries.

To address this need, we designed a randomised open-label clinical trial to determine the efficacy and safety of PA versus AL or DP, when administered to pregnant women with confirmed *P. falciparum* infection in the second or third trimester.

The specific trial objectives are:

1. To compare the efficacy of PA versus AL or DP in terms of (1) treatment success (see definition below) 28, 42 and 63 days after the start of treatment, with and without genotyping; (2) parasite clearance time, including submicroscopic malaria infections; (3) gametocyte carriage and clearance; (4) haematological recovery by Day 7, 14, 28, 42 and 63 after treatment and at delivery; (5) birth weight measured within 72 hours of delivery and (vi) placental *P. falciparum* malaria.
2. To describe the safety profile of PA, AL and DP in terms of (1) tolerability and (2) adverse events (AEs), including serious adverse events and adverse events of special interest, during the 63-day post-treatment follow-up (mother), at delivery (mother and baby), 1 month (mother and baby) and 1 year after the end of pregnancy (baby).
3. To explore the pharmacokinetics of pyronaridine in HIV-infected and non-HIV-infected pregnant women. This will be done on a small number of study subjects (60 women, 30 HIV uninfected and 30 HIV infected) in the two countries with the highest HIV prevalence, namely, the Democratic Republic of the Congo (DRC) and Mozambique.

## METHODS AND ANALYSIS
### Study setting
The trial will be implemented in five African countries with multiple sites of recruitment, as indicated in table 1.

The different sites were chosen to include African pregnant women who are residents of different malaria transmission facies, including the Gambia, which is in the pre-elimination phase.

| Table 1 | Countries and study site characteristics |
|---|---|
| Burkina-Faso: Nanoro | The study site is situated in the central-west of the country, 90 km from Ouagadougou in the Nanoro health district catchment area. Malaria transmission is high and seasonal. *Plasmodium falciparum* is the most common species of malaria parasite. *Anopheles gambiae* is the main vector transmitting the disease. |
| Democratic Republic of the Congo: Lisungi | The trial will be carried out at Lisungi Health Centre, located in Kinshasa suburb, where malaria transmission is perennial. *P. falciparum* is the most common species of malaria parasite. *Anopheles funestus* and *A. gambiae* are the main vectors transmitting the disease. |
| Mali: San and Téné | San Health Centre is located 440 km northeast of Bamako. Téné is about 50 km south of San. Malaria transmission is high and seasonal. *P. falciparum* is the most common species of malaria parasite. *A. gambiae* is the main vector transmitting the disease. |
| Mozambique: Manhiça District | Manhiça is located in southern Mozambique, 80 km north of Maputo city. Malaria is endemic with perennial transmission. *P. falciparum* is the most common species of malaria parasite. |
| The Gambia: Basse, Demba Kunta Koto, Fatoto, Gambissara and Koina | The trial will be carried out in the Upper River Region (eastern part of the country), where the MRCG has a field station. There are five health facilities (Basse, Demba Kunta Koto, Fatoto, Gambissara and Koina) from which pregnant women can be recruited. Malaria transmission is seasonal. *P. falciparum* is the most common species of malaria parasite. *A. gambiae* is the main vector transmitting the disease. *Anopheles arabiensis* and *A. funestus* are the main vectors transmitting the disease. |

The study (recruitment) started on February 2021 and the planned end date is June 2024.

## Patient and public involvement

Although patients were not involved in the design of the study, this project was designed to respond to their main concern, which is the reduction of malaria adverse effects in pregnant population. The governments of the five African countries were engaged through their National Malaria Control Programme (NMCPs), whose objective is to provide the evidence base to make an additional ACT available for malaria treatment in pregnancy. In the event of a positive result, it will be important to integrate this new treatment into (inter)national guidelines. This will be obtained by disseminating as rapidly and efficiently as possible the trial's results to relevant stakeholders, for example, the WHO, NMCPs and the scientific community working on malaria treatment. Through meetings in all the involved countries, permission from the community, religious leaders and women representatives has been sought before the trial starts. Therefore, similar channels will be used to share the study results. Participation in our trial can be burdensome as participants may endure psychological, cost and physical impacts. However, the project provided special attention such as closer supervision and treatment of any illness to enrolled patients. Also, transport costs have been provided to participants living far from the recruitment centre. Nonetheless, it has been suggested to all participants to withdraw their consent at any time if they feel burdened by the study procedures.

## Study design

This is an open-label, multicentre, randomised, non-inferiority clinical trial comparing PA with AL and DP for the treatment of *P. falciparum* malaria in women in the second and third trimesters of pregnancy. The non-inferiority design was suggested based on the hypothesis that PA, which is a newly registered ACT, is not worse than the other ACTs in terms of safety (for AL) and efficacy (for DP). The rationale for two control groups is that AL is currently the most used treatment for malaria in pregnancy, while DP seems to be the ACT with the best tolerability and efficacy profile.

The primary hypothesis is the pairwise non-inferiority of PA as compared with either AL or DP, defined as a difference in the Day 42 PCR-adjusted adequate clinical and parasitological response (ACPR) of <5% (non-inferiority margin).

## Participants and procedures

To be included, patients should satisfy all the inclusion criteria, while none of the exclusion criteria should be present.

### Inclusion criteria

1. Gestation ≥16 weeks and <37 weeks as assessed by ultrasound when possible. If not, the height of the uterus or delay of menstruation will be used.
2. *P. falciparum* monoinfection (by microscopy) of any density, regardless of symptoms and HIV status.
3. Haemoglobin ≥7 g/L.
4. Age ≥15 years.
5. Residence within the health facility catchment area.
6. Willingness to adhere to study requirements and to deliver at the health facility.
7. Ability to provide written informed consent; if the woman is a minor of age or not emancipated, the consent must be given by a parent or legal guardian according to national law (however, in this case, assent will be obtained to ensure that the woman herself is also freely willing to take part in the study).
8. For the PK study, HIV-infected women should be on first-line antiretroviral treatment for at least 6 months.

### Exclusion criteria

1. Known allergy to the study drugs.
2. History of known pregnancy complications or poor obstetric history, such as repeated stillbirths or eclampsia.
3. History or presence of major illnesses likely to influence pregnancy outcomes.
4. Any significant illness at the time of screening requiring hospitalisation, including:
   1. Severe malaria.
   2. Any sign or symptom suggesting hepatic lesions (eg, nausea with abdominal pain and icterus) or severe liver disease classified as B or C by the Child–Pugh score.
   3. Known history or evidence of clinically significant cardiovascular disorders or family history of long QT syndrome.
5. Intent to move out of the study catchment area before delivery or delivery out of the catchment area.
6. Prior enrolment in the study.
7. Clear evidence of recent (1 week) treatment with antimicrobials with antimalarial activity (azithromycin, clindamycin, tetracycline, quinolones, cotrimoxazole and SP). For HIV-infected pregnant women to be included in the PK substudy, cotrimoxazole use is not an exclusion criterion.
8. Twin/multiple pregnancy.
9. Known history of G6PD deficiency or sickle cell disease.

## Randomisation

### Sequence generation

Women recruited in the trial will be randomly allocated to one of the three treatment arms of the study according to a randomisation list generated using R software prior to any study activity.

The block randomisation technique will be used to achieve balance in the allocation of participants to treatment arms. A varying size of the blocks will be set up to reduce selection bias by using random block sizes and keeping the investigator blind to the size of each block, particularly in this open randomised control arm study. The randomisation will be done per study site within

each country and according to the specific sample size allocated to each site.

The data management team in each country, in coordination with the study statistician and the study data management centre, will print the randomisation code containing the study arm and put it into a sealed envelope numbered sequentially and containing the treatment arm to which the patient should be allocated.

## Recruitment

Women in the second or third trimester of pregnancy will be screened for malaria with a rapid diagnostic test based on the detection of histidine-rich protein 2 and parasite lactate; positive results will be confirmed by microscopy. Although malaria slides will be read by two certified microscopists, there is a small chance that mixed infections might be overlooked, and this can be considered a potential limitation of this study.

Women with a confirmed *P. falciparum* infection will be asked to provide written informed consent covering all trial procedures and be assigned a screening number. All women meeting the entry criteria will be given a randomisation number. Data from screened and randomised women will be kept in a logbook.

## Investigational products and comparators
### PA: Pyramax (Shin Poong Pharmaceutical Company, Korea)

PA is a film-coated tablet containing 180 mg of pyronaridine tetraphosphate and 60 mg of artesunate. The tablets should be taken orally once daily for 3 days and according to body weight as follows: 24 to <45 kg (two tablets), 45 to <65 kg (three tablets) and ≥65 kg (four tablets). This dosage regimen provides a daily dose of 7.2–13.8 mg/kg pyronaridine and 2.4–4.6 mg/kg artesunate. PA can be administered at any time, regardless of food consumption.

### DP (Alfasigma, Italy)

DP is a white film-coated tablet composed of 40 mg of dihydroartemisinin and 320 mg of piperaquine. DP tablets should be taken orally once daily for 3 days as follows: 24 to <36 kg (two tablets), 36 to <75 kg (three tablets) and ≥75 kg (four tablets).[9 10]

### AL: Coartem (Novartis)

AL tablets contain 80 mg of artemether and 480 mg of lumefantrine. This is a fixed-dose combination of artemether (a semi-synthetic artemisinin derivative) and lumefantrine (a slowly eliminated drug also referred to as benflumetol).[11] AL 80 mg/480 mg tablets should be taken orally two times per day for 3 days for patients weighing 35 kg and above:
► First dose, at the time of initial diagnosis (Day 0): one tablet.
► Second dose, at 8 hours after the first dose: one tablet.
► Third dose, in the morning of Day 1: one tablet.
► Fourth dose, in the evening of Day 1: one tablet.
► Fifth dose, in the morning of Day 2: one tablet.
► Sixth dose, in the evening of Day 2: one tablet.

Investigator's brochures and all documentation on drug quality will be provided by the manufacturers. The drugs will be shipped to the different sites with temperature monitors that record the temperature continuously. Once at the sites, the drugs will be stored and used according to the manufacturers' instructions. Independent study monitors (CliniPharm) will ensure compliance with good clinical practice, including good Investigational Product management practice.

## Explanation for the choice of comparators

Pregnant women (second or third trimester) will be recruited for the study and allocated to one of the three treatment groups. The choice of two control arms (AL and DP) is justified by AL being the most commonly used treatment for malaria in African pregnant women, while DP has several advantages compared with AL, namely higher efficacy and longer post-treatment prophylaxis.

## Study outcomes
### Primary endpoint

The primary endpoint is treatment efficacy, determined as the proportion of women with PCR-adjusted ACPR at Day 42, that is, all women not having met the criteria of treatment failure (online supplemental table 1). Recurrent infections classified by genotyping as new infections will not be considered treatment failures.

## Secondary endpoints

Safety is the main secondary endpoint and includes adverse events detected during active follow-up (63 days post-treatment), including significant changes in relevant laboratory values, those detected at delivery, at 4–6 weeks and 1 year after birth.

Other secondary endpoints include:
1. PCR-adjusted ACPR at Days 28 and 63.
2. PCR unadjusted ACPR on Days 28, 42 and 63.
3. Parasite and fever elimination time.
4. Gametocyte carriage and clearance.
5. Hematologic recovery, that is, haemoglobin changes between Day 0 and Days 7, 14, 28, 42, 63 and at delivery.
6. Placenta malaria (recent, past, and chronic infection).
7. Average birth weight and prevalence of LBW (<2500 g).

### Exploratory endpoints

We will explore (1) the drug exposure and key pharmacokinetic parameters of pyronaridine in HIV-uninfected pregnant women and (2) the drug exposure and key pharmacokinetic parameters of HIV-infected pregnant women on antiretroviral therapy. Drug exposure is defined as the area under the whole blood concentration versus the time curve from zero to infinity, $AUC_{0-\infty}$. The main pharmacokinetic parameters that will be evaluated are absorption rate ($k_a$), drug clearance (CL/F) and volume of distribution ($V_d/F$).

## Sample size

The sample size was estimated assuming an efficacy for PA (PCR adjusted at Day 42) of at least 95% and a non-inferiority margin of 5%; non-inferiority will be tested using raw pooling of country data using Wilson's interval of proportion difference. The lower limit of the Wilson score interval of 97.5% of the AL (or DP) proportion of ACPR – the PA proportion of ACPR – must be greater than –5% to claim non-inferiority. In addition, an adjusted non-inferiority analysis will allow for the country effect.[12] Assuming a 20% loss to follow-up, the total sample size for 90% power would be 3×500/0.8 = 1875, or 375 pregnant women per country and 125 per arm per country. The sample size calculated will maintain the power of the adjusted stratified non-inferiority test.

## Study implementation and timeline

Participants' assignment to the study arm is under the responsibility of study-qualified physicians. All physicians participating in the PYRAPREG study have been trained in good clinical practices and all study requirements.

Pregnant women fulfilling the inclusion/exclusion criteria will be recruited during antenatal clinics over a period of about 30 months. Scheduled visits will be on Days 3, 7 and then every week until Day 63 post-treatment. A window period is allowed if study subjects are unable to attend on the scheduled date, that is, ±1 day for Days 7 and 14; ±2 for Days 21 and 28; and±3 from Day 35 to Day 63. Women will be encouraged to attend the antenatal clinic between scheduled visits if they are sick.

Pregnant women recruited during the third trimester (before 37 weeks) may deliver before the end of the 63-day active follow-up. In this case, the assessment at delivery will be done as planned but the active follow-up will continue after delivery until Day 63. A blood sample to measure liver function tests (LFT) and bilirubin will be taken within 48 hours of delivery from babies whose mothers delivered within 2 weeks of their inclusion in the trial.

The outcome of pregnancy, birth weight and maternal haemoglobin will be collected as soon as possible after delivery. A placenta biopsy will be collected for later histopathological analysis. The newborn will be examined for congenital abnormalities. Both the mother and the newborn will be reassessed for any AEs between 4 and 6 weeks and then after 1 year (only the baby). Patients will be assessed as summarised in the study visit schedule (online supplemental table 2).

## Data collection and management

### Plans for assessment and collection of outcomes

#### Study visits

At each visit, both scheduled and unscheduled, the medical history since the last visit (including any treatment taken) and current signs and symptoms (if any) will be collected. A blood sample for a thick smear will be collected and the body temperature will be checked. Dried blood spots for later genotyping to distinguish between recrudescence and new infection will be collected at Day 0, before treatment, and at every study visit. Information on any AE will also be collected. Haematology (full blood count) will be performed at Days 0, 7, 14, 28, 42 and 63; biochemistry (total and conjugated bilirubin, aspartate transaminase (AST), alanine transaminase (ALT), alkaline phosphatase (ALP) and creatinine) at Days 0, 1 and 7. In the event of increased LFTs, >3× upper limit normal (ULN), the result will be verified by taking an additional sample to be analysed (within 24 hours). Subsequent blood samples for LFTs will be taken at 48-hour intervals until the results return to <2× ULN. ECG will be performed on Day 0, before drug intake, and on Day 2, after drug intake. If abnormal on Day 2, the ECG will be repeated on Day 7 and every week until returns to normal. At the end of the active follow-up, on Day 63, field assistants will visit the study subjects monthly to maintain contact, but without collecting any data or biological samples.

Babies born from women who deliver within or 2 weeks after the active follow-up will have a blood sample taken to measure LFT and bilirubin within 48 hours of delivery.

The outcome of pregnancy, birth weight and maternal haemoglobin will be collected as soon as possible after delivery. A placenta biopsy will be collected and put immediately in a 10% buffered formalin container stored at 4°C at the study site for later histopathological analysis. The newborn will be examined for congenital abnormalities. Both the mother and the newborn will be reassessed twice after delivery for any AE: between 4 and 6 weeks and then after 1 year.

### PCR analysis

Genotyping of recurrent infections will be performed by characterising the merozoites surface protein 1 (*msp1*), *msp2* and glutamate-rich protein (*glurp*) genes, single-copy genes of the *P. falciparum* genome. PCR amplification of DNA from a single parasite clone will yield a unique amplification product. For all three genes, each PCR amplification product of a different size is considered from a different *P. falciparum* clone and reflects a different genotype. For samples collected from the same patient on Day 0 and on the day of recurrent parasitaemia (after Day 3), the length polymorphism of *msp1*, *msp2* and *glurp* will be determined, that is, the number of bands in each PCR reaction and their respective sizes. The results will be interpreted as follows:

► *Recurrence*: at least one polymorphism of identical length for each marker (*msp1*, *msp2* and *glurp*) is found in the sample collected on Day 0 and on the day of the recurrent parasitaemia.
► *New infection*: for at least one marker, the length polymorphism is different between the sample collected on Day 0 and on the day of recurrent parasitaemia.
► *Indeterminate*: samples that did not give a result due to an inability to amplify DNA on Day 0 and/or on the day of recurrent parasitaemia.

## Analysis of placental samples

The placental biopsy samples will be processed and embedded in paraffin wax using standard techniques. Paraffin sections 4 μm thick will be stained with H&E. Placental biopsies after reading will be classified according to the following definitions[13]: (1) acute infection (parasite present, haemozoin absent or minimal deposition); (2) chronic infection (parasites and heavy haemozoin deposition); (3) past infection (no parasite but presence of haemozoin) or (4) no infection (absence of parasites and haemozoin).

## Haematological and biochemical analysis

Haematology (including haemoglobin) and biochemistry (including LFTs, that is, AST, ALT, ALP, total and conjugated bilirubin) will be performed during active follow-up; haematology will be performed prior to the first dose of treatment, on Day 0, and then on Days 7, 14, 28, 42 and 63. An additional test will be performed at any unscheduled visit. In addition, only haemoglobin will be measured at delivery. Biochemistry will also be done on Day 0, prior to treatment, then on Day 1 and Day 7. If LFTs increase more than three times the ULN, the result will be verified by taking another sample for analysis as soon as possible (within 24 hours of the initial sample). Subsequent blood samples for LFTs will be taken at 48-hour intervals until results return to <2× ULN.

## Statistical methods

### Statistical methods for primary and secondary outcomes

The baseline characteristics will be described by treatment group and site.

The primary analysis will be the assessment of the non-inferiority of the PA compared with the DP and AL for the PCR-adjusted ACPR at Day 42. It will use the combined data from the five countries, with adjustment for any centre effect, using an additive model for response rates (ie, a generalised linear model with a Bernoulli error distribution and an identity link function). This will allow the evaluation of two pairwise treatment comparisons, that is, PA versus AL and PA versus DP.

### Efficacy analysis (primary endpoint)

For the efficacy analysis, both a modified-intention to treat (m-ITT) approach and a per-protocol (PP) approach will be adopted, with PP analysis being the main approach, as recommended for equivalence studies. The m-ITT population will include all participants who have received any amount of the study drug and have confirmed *P. falciparum* infection prior to treatment.

The PP population will consist of all participants meeting the following predefined criteria:
1. Fulfilling the entry criteria specified in the clinical study protocol.
2. Completed treatment, including not having vomited the study drug or, if vomited, receiving a repeat dose that was not vomited.

3. No previous or concomitant medication would interfere with the treatment outcome.

The PP population will be identified after locking the database, just before the statistical analysis.

For the primary endpoint, that is, treatment efficacy on Day 42, the proportion of participants with PCR-adjusted ACPR will be determined by the treatment arm. Similar procedures will be applied to the m-ITT population.

### Safety analysis (secondary endpoint)

The safety population will include all participants randomised and treated with at least one dose of the three antimalarial treatments. Standard safety report tables summarise and list safety data. All AEs will be coded using the most recent version of the Medical Dictionary for Regulatory Activities (MedDRA) dictionary. Treatment-emergent AEs will be defined as all AEs that started after the first administration of the study drug. These AEs will be summarised by primary system organ class and preferred term, separately for each treatment regimen and overall. Similar summaries will be provided for treatment-emergent AEs considered to be related to study treatments. In addition, treatment-emergent AEs will be summarised by primary system organ class, preferred term, and maximal severity.

Vital signs and routine safety laboratory data will be summarised descriptively by treatment regimen and overall, by time point. Absolute values and changes from the baseline will be presented. Safety laboratory data will be classified according to the normal ranges (below, within and above), and summaries of changes from baseline in these categories will be provided by treatment regimen and overall. Furthermore, safety laboratory values will be classified according to Common Terminology Criteria for Adverse Events (CTCAE) and shift tables of the baseline CTCAE category versus post-baseline categories will be presented.

### Pharmacokinetic analysis

Pyronaridine exposure in patients' blood will be assessed in pregnant women, both infected and non-infected by HIV, by determining pyronaridine concentrations in EDTA whole blood samples collected on Day 0 before and 1 hour, 2 hours, 6 hours and 10 hours after the first PA dose, on Day 1 (24 hours after the first PA dose while before the second PA dose), Day 2 (before the last PA dose), and subsequently on Days 7, 14, 21, 28 and 42 after initiation of treatment. A population pharmacokinetic approach using non-linear mixed effects modeling will be employed to analyze the pharmacokinetic data. A compartmental population pharmacokinetic model will be developed, including a stochastic model describing between-subject and residual variability. In short, the main parameter of interest is overall whole blood pyronaridine exposure, defined as the area under the concentration-time curve until infinity ($AUC_{0-inf}$), which will be characterised based on the final individual model parameter estimates and compared with a separate

cohort of non-pregnant malaria patients treated with PA. The effect of pregnancy will be evaluated and quantified either as a binary or continuous (estimated gestational age) covariate on all primary estimated pharmacokinetic parameters (such as $K_a$, $\mathrm{CL}/F$ and $V_d/F$). Furthermore, we will explore the effect of HIV background (disease effects and potential antiretroviral drug–drug interactions) on the pharmacokinetics of pyronaridine by evaluating HIV status as a covariate. Statistical significance and selection of covariates in nested population pharmacokinetic models will be based on a likelihood ratio test to evaluate the difference in model fit.

## Study monitoring

The trial will be evaluated by monitors (a Clinical Research Organisation will be contracted) (prestudy visit) for its preparedness to carry out, followed by regular monitoring and closeout visits.

The coordination of the whole project is in the hands of the sponsor, who will be the primary contact. He will be assisted by appropriate administrative and financial staff. A Coordinating Committee (CC), including one member of each institution, will be the main decision body of the consortium. For the trial, there will be three entities involved in its implementation and management, namely, the Data Safety and Monitoring Board (DSMB), the Trial Steering Committee (TSC), and the Trial Management Group. Both DSMB and TSC are composed of independent experts to provide the overall supervision of the trial, monitor trial progress, and advise on scientific credibility.

## ETHICS AND DISSEMINATION

This protocol has been approved by the Ethics Committee for Health Research (CERS) in Burkina Faso (Reference: 2020-3-047), the National Health Ethics Committee (CNES) in the Democratic Republic of Congo (Reference: 169/CNES/BN/PMMF/2019), the Ethics Committee of the Faculty of Medicine and Odontostomatology (FMOS)/Faculty of Pharmacy (FPHA) in Mali (Reference: 2020/46/CE/FMOS/FAPH), the Gambia Government/MRCG Joint Ethics Committee (Reference: 21818) and the National Bioethics Committee for Health (CNBS) in Mozambique (Reference: 313/CNBS/20).

Written informed consent will be obtained from each individual prior to her participation in the study by the study investigators.

The outcomes of the project will be communicated to the NMCP of the respective countries and to the Global Malaria Programme of the WHO. The results will be published in peer-reviewed open access journals and presented at (inter)national conferences and meetings.

A work package is focused on the development of a plan for internal and external project communication, the development of communication tools, and the dissemination and exploitation of the project's findings. All these activities will be in line with H2020 guidelines on dissemination and publication of results and will highlight the contribution of the EDCTP in tackling societal and health challenges.

## Trial status

Active recruitment.

## Trial sponsor and role

The trial sponsor is the University of Sciences, Techniques, and Technologies of Bamako (USTTB). Hamdallaye ACI 2000, Rue 405 - Porte 359, Bamako

Téléphone : +223 20 29 04 07

http://usttb.edu.ml/

The trial sponsor in the study has the responsibility to ensure that data and source documents are granted for trial monitoring, audits, DSMB and Ethics Committee reviews, and regulatory inspections as appropriate.

**Author affiliations**
[1]Malaria Research and Training Center, University of Sciences Techniques and Technologies of Bamako, Bamako, Mali
[2]Département of Tropical Médecine, Universite de Kinshasa, Kinshasa, Democratic Republic of Congo
[3]Department of Tropical Medicine, Universite de Kinshasa Faculte de Medecine, Kinshasa, Democratic Republic of Congo
[4]Institut de Recherche en Sciences de la Santé (IRSS) – Unité de Recherche Clinique de Nanoro, Nanoro, Burkina Faso
[5]Centro de Investigacao em Saude de Manhica, Manhica, Mozambique
[6]Universidade Eduardo Mondlane, Maputo, Mozambique
[7]Institut de Recherche en Sciences de la Santé (IRSS) – Unité de Recherche Clinique de Nanoro, Ouagadougou, Burkina Faso
[8]Centro de Investigação em Saúde de Manhiça, Manhica, Mozambique
[9]Centro de Investigação em Saúde da Manhiça (CISM), Maputo, Mozambique
[10]MRC Unit The Gambia (MRCG) at the London School of Hygiene and Tropical Medicine, The Gambia London, UK
[11]Faculty of Medicine Odontostomatology, University of Sciences Techniques and Technologies of Bamako, Bamako, Mali
[12]Instituto de Salud Global Barcelona, Barcelona, Spain
[13]Netherlands Cancer Institute / Antoni van Leeuwenhoek Hospital, Amsterdam, The Netherlands
[14]Department of Pharmacy, Uppsala University, Uppsala, UK
[15]Amsterdam University Medical Centres, Academic Medical Centre at the University of Amsterdam (AMC), Amsterdam, The Netherlands
[16]MRC Laboratories The Gambia, Banjul, Gambia

**Acknowledgements** The consortium thanks Shinpoong Pharm.Co., Ltd. for providing the study drug (pyronaridine-artesunate: Pyramax) and the Medicine of Malaria Venture (MMV) for technical support and assistance in this ongoing clinical trial. The project would like to thank patients and their advisors for their contribution to the study's implementation.

**Contributors** The following co-authors were responsible for the conception and design of the clinical trial: KK, UD, HMM, HT, MT, ES, MP, RG, CM, TPCD, PFM and HDS. The first draft of the paper was written by MD and JKT. IS is responsible for the statistical analysis. All authors and the following (AV, SM, BK, EDD, AE, HD and MK) contributed to critical review and approved the final manuscript.

**Funding** This project is part of the EDCTP2 programme supported by the European Union (grant number RIA2017MC-2025-PYRAPREG).

**Competing interests** None declared.

**Patient and public involvement** Patients and/or the public were not involved in the design, or conduct, or reporting, or dissemination plans of this research.

**Patient consent for publication** Consent obtained directly from patient(s).

**Provenance and peer review** Not commissioned; externally peer reviewed.

**ORCID iDs**
Japhet Kabalu Tshiongo http://orcid.org/0000-0002-2344-1111
Raquel González http://orcid.org/0000-0001-5487-801X
Kassoum Kayentao http://orcid.org/0000-0001-6877-0093

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
