## [Reviewer comments · BMJ Open]

ARTICLE DETAILS

TITLE (PROVISIONAL)	Efficacy and safety of pyronaridine-artesunate (PYRAMAX [®]) for the treatment of P. falciparum uncomplicated malaria in African pregnant women (PYRAPREG): study protocol for a phase 3, non-inferiority, randomized open-label clinical trial
AUTHORS	Djimde, Moussa; Tshiongo, Japhet; Muhindo, Hypolite; Tinto, Halidou; Sevene, Esperanca; Traore, Maminata; Vala, Anifa; Macuacua, Salesio; Kabore, Berenger; Dabira, Edgard; Erhart, Annette; Diakite, Hamadoun; Keita, Mohamed; Piqueras, Mireia; González, Raquel; Menendez, Clara; Dorlo, Thomas; Sagara, Issaka; Mens, Petra; Schallig, Henk; D'Alessandro, Umberto; Kayentao, Kassoum

VERSION 1 – REVIEW

REVIEWER	Vineeta Singh ICMR-National Institute of Malaria Research
REVIEW RETURNED	06-Feb-2023

GENERAL COMMENTS	Manuscript ID: bmjopen-2022-065295 Title: "Efficacyandsafetyofpyronaridine-artesunate(PYRAMAX[®])forthetreatmentofPlasmodium falciparum malaria in African pregnant women (PYRAPREG): study protocol for a phase 3, non- inferiority, randomized open-label clinical trial." Authors: Djimde et al. I read with interest protocol paper by Djimde and colleagues on efficacy and safety of pyronaridine + artesunate, a recently WHO endorsed ACTs. This protocol is well structured and future results will be be very helpful to know if this ACT could be used as treatment of P. falciparum uncomplicated malaria during pregnancy. However I have some comments that you will find below Comments Title 1) Based on your inclusion/exclusion criteria you should revise title by adding the term “uncomplicated” Abstract (Page 5 of 46) 2) Lines 21 – 26: One verb is missing in this sentence. Please check it 3) Lines 50 – 54: Please rearrange keywords. Keep in your mind that keywords should be ordered by decreasing order of their importance. For instance, Pregnant women should be placed directed after Malaria. Strengths and Limitations (Page 6 of 46) 4) Line 7: Change “(Pyronaridine-artesunate (PA))” to “Pyronaridine – artesunate (PA)”. Introduction (Page 7 of 46) 5) Lines 6 – 8: The sentence “due to immunological and hormonal changes” is not specific to explain why malaria risk is higher in pregnant women. These changes also happen during adolescence, elderly
--

	people and menopause. So, I suggest to specifically state that pregnancy is associated with hormonal changes and reduced immunity that expose pregnant women to malaria. 6) Lines 13 – 16: You outlined that effective case management of clinical malaria relies on ACTs. This statement is misleading as ACTs are recommended for P. falciparum infection during second and third trimesters of pregnancy while Quinine + Clindamycin for 1st trimester. For P. vivax malaria, Quinine or chloroquine are used during first trimester. You can check this recent review paper (https://www.tandfonline.com/doi/full/10.1080/20477724.2022.2128563), and so, rephrase these sentences properly 7) Line 35: Change “need of” to “need for” 8) Line 37: Change “for treating” to “to treat” 9) Line 49: on this line you said “efficacy, safety and tolerability”, but in previous sections you just said “efficacy and safety” (e.g., Page 6 of 46, line 12). Please adjust it 10) Line 3 (Page 8 of 46): Change “Birth” to “birth” Methods and Analysis 11) Table 1 (Page 8 of 46): You used different adjective to characterize malaria transmission (e.g., intense, high, perennial, etc.) Note that there is formal categorization to define level of malaria transmission. Please standardize it as per WHO definitions. Also, for some you talked about intensity and seasonality of malaria transmission, for some you just talked about either seasonality or intensity. Please uniform it! 12) Table 1 (Page 8 of 46): Please add information about climatic characteristics, main vegetation, Anopheles vectors, and which malaria species are present for each site. 13) You have to justify why you selected these countries. Noted that four of African countries you selected are targeted by HBHI strategy, with exception of The Gambia. So, why you selected “The Gambia”? 14) You have to justify why you designed your trial as “non-inferiority” 15) Line 14 (Page 10 of 16): I don’t think mono-infection based on microscopy is a good option, especially when other malaria species circulates. So, you can mistakenly include mixed infections 16) Lines 24 – 28 (Page 10 of 46): The recent HELSINKI international recommendations ask to also take consent from minors along with theirs of parents/guardians. So, you also have to ask written informed consent for minors. Please add it 17) Lines 43-44 (Page 11 of 46): Please specify which type of RDTs will be used as per WHO categorization 18) How and where will you afford different drugs? Clearly specify it that there might some conflicts of interests from few authors of this protocol 19) How will you ascertain quality of drugs? 20) You know that you should use comparative groups, and these groups should be similar for main characteristics such as age and parity. So, how will you guarantee that your groups will be similar? Statistical methods 21) You have to do comparative analysis between groups for main sociodemographic and clinical characteristics using Chi-square, t test or their non-parametric alternatives. 22) Lines 50-57 (Page 17 of 46): Clearly specify which methods you will use to analyze bio-availability Abbreviations
--	--

	Some abbreviations were used just once, so you don't need to present them in this section. As a consequence, you removed these abbreviations in the text. Other abbreviations have not been ever used in the text but you presented in this section (e.g., LLIN). So, adjust text and this section. References This section should be properly presented as per BMJ guidelines. For instance, for some reference information such as volume, issue and page numbers are missing (e.g. Ref N° 15, 16) while for other this information are present (e.g., Ref N° 18). Check carefully all references one by one to meet BMJ guidelines.
--	---

REVIEWER	Richard Kajubi Infectious Diseases Research Collaboration
REVIEW RETURNED	03-Mar-2023

GENERAL COMMENTS	I think the study is well-designed to answer pertinent questions on the efficacy and safety of an additional ACT for use in pregnancy. Generally, the paper is well written though the grammatically it could still be improved.
--

VERSION 1 – AUTHOR RESPONSE

Reviewer1

1) Based on your inclusion/exclusion criteria you should revise title by adding the term "uncomplicated"

Response: The term "uncomplicated" has been added to the current title as suggested.

Abstract

2) Lines 21 – 26: One verb is missing in this sentence. Please check it

Response: Thank you for the observation; We have checked the said sentence on lines 21-26 and replaced the verb "taken" by "obtained".

3) Lines 50 – 54: Please rearrange keywords. Keep in your mind that keywords should be ordered by decreasing order of their importance. For instance, Pregnant women should be placed directed after Malaria.

Response: As advised by the reviewer, we have rearranged the order of the keywords by placing "pregnant women" just after "malaria".

Strengths and Limitations

4) Line 7: Change "(Pyronaridine-artesunate (PA))" to "Pyronaridine – artesunate (PA)".

Introduction (Page 7 of 46)

Response: We have replaced "Pyronaridine-artesunate (PA)" to "pyronaridine - artesunate (PA)".

Introduction

5) Lines 6 – 8: The sentence "due to immunological and hormonal changes" is not specific to explain why malaria risk is higher in pregnant women. These changes also happen during adolescence, elderly people and menopause. So, I suggest to specifically state that pregnancy is associated with hormonal changes and reduced immunity that expose pregnant women to malaria.

Response: As advised, we have reworded the first sentence of the introduction as “Pregnancy is associated with hormonal changes and reduced immunity that make pregnant women more vulnerable malaria”.

6) Lines 13 – 16: You outlined that effective case management of clinical malaria relies on ACTs. This statement is misleading as ACTs are recommended for *P. falciparum* infection during second and third trimesters of pregnancy while Quinine + Clindamycin for 1st trimester. For *P. vivax* malaria, Quinine or chloroquine are used during first trimester. You can check this recent review paper
Response: The sentence on lines 13 - 16 has been reworded correctly as suggested.

7) Line 35: Change “need of” to “need for”

Response: As suggested, we replaced on line 35 “need of” by “need for”.

8) Line 37: Change “for treating” to “to treat”

Response: As suggested, we replaced “for treating” by “to treat”.

9) Line 49: on this line you said “efficacy, safety and tolerability”, but in previous sections you just said “efficacy and safety” (e.g., Page 6 of 46, line 12). Please adjust it

Response: The said sentence has been adjusted by removing tolerability.

10) Line 3 (Page 8 of 46): Change “Birth” to “birth”

Response: As recommended, we changed “Birth” to “birth”.

Methods and Analysis

11) Table 1 (Page 8 of 46): You used different adjective to characterize malaria transmission (e.g., intense, high, perennial, etc.) Note that there is formal categorization to define level of malaria transmission. Please standardize it as per WHO definitions. Also, for some you talked about intensity and seasonality of malaria transmission, for some you just talked about either seasonality or intensity. Please uniform it!

Response: The information from the different sites has been standardized.

12) Table 1 (Page 8 of 46): Please add information about climatic characteristics, main vegetation, Anopheles vectors, and which malaria species are present for each site.

Response: More details have been provided on the description of the Study Sites.

13) You have to justify why you selected these countries. Noted that four of African countries you selected are targeted by HBHI strategy, with exception of The Gambia. So, why you selected “The Gambia”?

Response: We agree with the reviewer about the decrease of malaria in the Gambia. However, the rationale for the choice of sites including The Gambia has been added to the manuscript. The Gambia is of interest because it is in pre-elimination stage.

14) You have to justify why you designed your trial as “non-inferiority”

Response: The non-inferiority design was suggested based on the hypothesis that PA, which is a newly registered ACT is not worse than the other ACT in terms of safety (for AL) and efficacy (for DP). The rationale for 2 control groups is that AL is currently the most used treatment for malaria in pregnancy while DP seems to be the ACT with the best tolerability and efficacy profile. We have added this in the justification.

15)) Line 14 (Page 10 of 16): I don't think mono-infection based on microscopy is a good option, especially when other malaria species circulates. So, you can mistakenly include mixed infections.

Response: We recognize the limitation of microscopy; however, the reading is done by two certified microscopists.

16) Lines 24 – 28 (Page 10 of 46): The recent HELSINKI international recommendations ask to also take consent from minors along with theirs of parents/guardians. So, you also have to ask written informed consent for minors. Please add it

Response: We have added the suggestion that for pregnant minors, assent will be obtained to ensures that the woman herself is also freely willing to take part in the study.

17) Lines 43-44 (Page 11 of 46): Please specify which type of RDTs will be used as per WHO categorization

Response: We have added the specification of type of RDTs which will be used.

18) How and where will you afford different drugs? Clearly specify it that there might some conflicts of interests from few authors of this protocol.

Response: All authors of this publication declared not to have a conflict of interest with any of the companies supplying the study drugs.

19) How will you ascertain quality of drugs?

Response: Investigator's brochures and all documentation on drug quality will be provided by the manufacturers. The drugs will be shipped to the different sites with temperature monitors recording the temperature continuously. Once at the sites, the drugs will be stored and used according to the manufacturers' instructions. Independent study monitors (CliniPharm) will ensure compliance with good clinical practice including good Investigational Product management practice.

20) You know that you should use comparative groups, and these groups should be similar for main characteristics such as age and parity. So, how will you guarantee that your groups will be similar?

Statistical methods

Response: Randomization will ensure that characteristics such as age and parity will be similar among groups.

Statistical methods

21) You have to do comparative analysis between groups for main sociodemographic and clinical characteristics using Chi-square, t test or their non-parametric alternatives.

Response: To compare groups for main sociodemographic and clinical characteristics we will stick to the conditions of application of the tests. However, with our large sample size, we believe we can use parametric tests.

22) Lines 50-57 (Page 17 of 46): Clearly specify which methods you will use to analyze bio-availability Abbreviations Some abbreviations were used just once, so you don't need to present them in this section. As a consequence, you removed these abbreviations in the text. Other abbreviations have not been ever used in the text but you presented in this section (e.g., LLIN). So, adjust text and this section.

Response: We added the sentence the following sentence "Mean of PA concentration will be estimated and compared between pregnant and non-pregnant women over time. Similar figure will be used to compared HIV infected and non-infected pregnant women".

References

23) This section should be properly presented as per BMJ guidelines. For instance, for some reference information such as volume, issue and page numbers are missing (e.g. Ref N° 15, 16) while

for other this information are present (e.g., Ref N° 18). Check carefully all references one by one to meet BMJ guidelines.

Response: All references have been updated and adapted to the BMJ Open style.

Reviewer 2

I think the study is well-designed to answer pertinent questions on the efficacy and safety of an additional ACT for use in pregnancy. Generally, the paper is well written though the grammatically it could still be improved.

Response: Thank you for your kind support